# Return Strategies and Online Product Customization in a Dual-Channel Supply Chain

**Rong Zhang [1], Jiatong Li [2], Zongsheng Huang [2] and Bin Liu [2,*]**

[1]  Research Center of Logistics, Shanghai Maritime University, Shanghai 201306, China;
   Zhangrong@shmtu.edu.cn
[2]  School of Economics and Management, Shanghai Maritime University, Shanghai 201306, China;
   Lijiatong045@foxmail.com (J.L.); zshuang@shmtu.edu.cn (Z.H.)
*  Correspondence: Liubin@shmtu.edu.cn

**Abstract:** This paper investigates in a dual-channel supply chain which return strategy is better for the manufacturer that considers the consumers' utility. We find that a manufacturer prefers offering a Money-Back Guarantee (MBG) as long as the net salvage value of the returned product is positive in a channel. However, the return strategy of the retailer is more affected by the return policy of another channel than the net salvage value. In order to reduce online returns, we propose the online product customization channel, and then, we examine the choice of return policy and the manufacturer's channel selection. We show that the demand and profit of the manufacturer will increase to a certain extent when opening an online customization channel. However, compared to the case where both channels provide an MBG, the implementation of online customization may hurt the manufacturer's profits with the increase in consumer satisfaction in indirect channels.

**Keywords:** return policy; online customization; pricing; Stackelberg game

## 1. Introduction

Manufacturers are more willing to open a direct sales channel after a traditional sales channel already exists. They prefer to sell products directly to customers in order to improve market share. It is easier to establish online direct sales channels due to the growth of network technology, and the convenience of online shopping attracts a large number of consumers [1]. More and more consumers buy products moving from the offline to the online channel. A study by Forrester Research noted that online sales channel of the manufacturer would account for 34% of total sales by 2016 [2]. According to the report of the National Bureau of Statistics of China, the online retail sales in China reached nine trillion yuan in 2018, an increase of 23.9% over the previous year, accounting for 18.4% of the total retail sales of consumer goods. Compared with traditional brick-and-mortar retail channels, the online channel can capture customers that value the convenience of shopping and save costs for manufacturers [3].

When consumers purchase products, they will consider the services provided by the companies, except the performance of the product itself. The return policy is quite an important service. In practice, customers in numerous industries have the right to return products to enterprises [4], and the online channels are no exception. Customers choose a product only relying on browsing product descriptions and images on the website. Due to the asymmetry of commodity information, it is difficult for customers to evaluate products accurately in e-commerce. Customers have "touch and feel" experiences by shopping through the traditional retail channel [5], which is hard to achieve in online channels. This ex-ante uncertainty brings about the conflict between commodities and customers' preference, which often leads to the return of products. When a company provides a Money-Back Guarantee (MBG)

return policy, consumers can choose to return the non-suitable product within a limited time. Since there is no quality problem with this kind of returned product, re-selling or recycling as a component can save costs and resources. Therefore, the manufacturer's MBG return policy not only provides consumers with a guarantee, but it also contributes to resource conservation. When making purchase decisions, over 74% of customers first consider after-sales service and return policy, according to a survey [6]. Toktay P pointed out that the return rate reached 5–9% of sales, while the online return rate was much higher than this [7]. The National Retail Federation (2014) pointed out that the return rates of online channels are typically between 20% and 40%, with poor-fit cited as the main reason. In addition, 72% of enterprises cover the cost of handling returns [8], which makes returns a significant part of the company's cost. According to the Center for Logistics Management at the University of Nevada, Reno, returned products cost the top 30 non-grocery retailers some $53 billion per year. Returns are often regarded as the core weakness of online channels [5]. Nonetheless, a growing number of manufacturers now prefer setting up online channels based on traditional sales channels. For example, an information technology enterprise like Apple currently operates 500 retail stores in the world and operates the online Apple Store, which provides MBG under broad circumstances.

To improve consumers' satisfaction with online shopping, reduce the high return rates caused by the poor-fit cited, and stimulate online customization, some manufacturers started to sell personalized products through the online channel to make full use of the advantages of the online channel. Online customization has become a new choice of many consumers in online customization behavior because of the diversity of product attribute selections and flexibility of customization time [9]. For example, Nike launched a customization platform to let consumers be the designers of their own shoes as early as 2008. Consumers can choose a Nike standard sneaker at will and match the color, material, and other elements provided by the platform to make a DIY pair of shoes. Apple's online store offers a personalized laser engraving service for a variety of products, and Apple Watch has provided 34 different online product combinations to meet the specific preferences of different consumers.

Research about the dual-channel supply chain has attracted a large number of scholars, and the main research areas include price decision, channel conflict, and channel coordination. However, most studies did not take into account returns, which is an important aspect of the dual-channel supply chain. Returns are often viewed as a cost center of companies [10]. McWilliams (2012) found that a Money Back Guarantee (MBG) is ubiquitous among major retailers [11]. In dual-channel structures; however, if the return rate of the online channel is high, should the manufacturer choose a full-refund policy for the channel?

If the high return rates of the online channel hurt enterprises, one of the potential advantages of the online channel is that it is easy to implement product customization. Online customization in this paper means that consumers select, adjust, or create a combination of options for a product through the online customization toolkit provided by an enterprise or an online retailer that meets their customization needs and unique preferences. Consumers distinguish themselves from the masses by showing and using the product [12,13]. Online customization emphasizes the uniqueness and differentiation of customers, and the customized content is concentrated on the appearance of the product. Due to the specificity of customized products, most manufacturers that offer customized services will not provide a return policy (except quality issues). It remains to be seen whether an online customization channel will bring more benefits than a regular online channel.

In this paper, we develop a dual-supply chain model to find the optimal return strategy and pricing for the manufacturer and retailer, and we introduce the implementation of personalized products in the online channel to analyze the change of return policy and pricing decisions. In addition, we also identify conditions under which the manufacturer should choose an online customization channel by numerical examples. Through the analysis, we show that the manufacturer should provide an MBG for any one (or both) of the channels when the net salvage value of returned products in the channel is positive, while retailers prefer to reduce competition between channels rather than opening an MBG. We also find that the online demand and profit of the manufacturer will increase to a certain extent

when opening online customization channels. With the increase in consumer satisfaction in indirect channels, the implementation of online customization will hurt the manufacturer's profits.

The remainder of this paper is organized as follows. Section 2 reviews relevant literature. In Section 3, we set up a dual-channel model that considers the return policy, obtain the optimal price, and discuss the optimal return strategy of the dual-channel structure in Section 4. Section 5 introduces product customization in online channels and analyzes the impact on price decisions. We also identify the conditions of opening an online customization channel in Section 5. In Section 6, we offer some concluding remarks and directions for future research. All proofs are in Appendix A.

## 2. Literature Review

This section provides an overview of the literature on the dual-channel supply chain, customer return strategies, and online customization.

The dual-channel supply chain has been extensively studied in the literature. Chiang et al. (2003) pointed out that direct channels can indirectly increase the profits of traditional retail channels and found that manufacturers can use direct channels to control retailers' prices, thus reducing the double marginalization issue [14]. Yan et al. (2009) pointed out that opening a direct channel is not always unfavorable for retailers, due to the lower wholesale price from the manufacturer, and is also promoting retailers to improve service quality [15]. Yoo and Lee (2011) studied the impact of online channel entry in the specific channel structure and varying market conditions, showing that the online channel does not always lead to lower retail prices and increased consumer welfare [16]. Soysal and Krishnamurthi (2015) analyzed the impact of introducing low-quality direct sales channels on high-quality channels' profitability [17]. They found that opening a low-quality, low-price direct sales channel can significantly increase the quantity of retail channel purchases. Erjiang et al. (2016) studied cooperative promotion in competing supply chains and found that an appropriate cost sharing construct can benefit both manufacturers and retailers [18].

It is common in the retail industry for customers to return products they have purchased. While accepting and handling returned products can reduce companies' profits, the service can improve customer satisfaction and lead to higher demand. Baiman et al. (2000) showed that an unconditional full return policy will give consumers a sense of trust in the product quality, but it will also increase unnecessary returns [19]. Mukhopadhyay and Setoputro (2004) determined the necessity of offering a return policy in direct channels and obtained the optimal return policy through the profit maximization model [6]. McWilliams (2012) explored the impact of MBGs on the profits of two competing retailers and found that an MBG not only does not disadvantage low-quality enterprises, but also improves their profit level [11]. The above paper focused on direct channel return policies, while many manufacturers provide return policies mainly for retail channels [20]. Davis (2001) proposed that even though the number of e-retailers providing return policies is growing, brick-and-mortar retailers are still leading the trend of return policies [21]. Chen and Bell (2012) found that dual-channel systems that allow returns and do not allow returns can increase retailers' profits, but they do not consider retailers' return policy options [3]. Shi and Xiao (2015) constructed a game model of supplier-managed inventory to analyze the impact of both supply chain decentralization and the service subsidy rate on the return policy and found that if the market is small enough, returns will not be allowed [22]. However, a few studies focused on how return policies affect retail price and order quantity and disregarded which return strategy was optimal for companies in the dual-channel supply chain. Our paper introduces a two-stage Stackelberg model that allows the manufacturer to offer an MBG return policy in an online or indirect channel and implement customization in the online channel.

Our paper is also related to online customization. Franke and Schreier (2010) pointed out that consumer preference is an attitude that guides consumers in purchasing decision-making directions and behaviors [23]. Kramer et al. (2007) pointed out that customization means that a company can offer personalized products to meet the needs and preferences of different consumers [24]. Previous research has focused on mass customization. This is different from mass customization. Jiang et al.

(2011) identified that mass customization emphasizes meeting the specific needs of consumers in terms of functionality, and the customizable content is focused on the intrinsic functional attributes of the product [25]. Online customization focuses more on the appearance of products. The purpose is to emphasize product specificity while highlighting the uniqueness of consumers [26,27]. Driven by e-commerce, online customization technologies are growing rapidly. With an online product customization toolkit, firms can offer different product attributes and features for customers to match personal preferences [28,29]. To the best of our knowledge, the study of online customization in dual-channel settings has been limited. Meanwhile, none of these studies explored the interaction between online product customization and returns.

## 3. Dual-Channel Model

We start by considering a manufacturer-led dual-channel model, in which a manufacturer sells a general product directly to consumers and indirectly through the retailer. We define the two channels as the direct channel (i.e., the online channel, $j = m$) and indirect channel (i.e., the traditional retail channel, $j = r$). The manufacturer decides the retail price in channel $m$, which is denoted by $p_m^K$; while in the indirect channel, the manufacturer offers the wholesale price $w^K$ to the retailer, and then, the retailer decides the final retail price $p_r^K$. Apart from the pricing decision, the two members of the supply chain can also decide return policy $k_j$ for either channel $j$: either MBG ($k_j = G$) or no MBG ($k_j = N$). Therefore, we consider four return strategies: $K = k_m k_r = \{NN, NG, GN, GG\}$. In practice, companies set the return policies to provide post-sales service to satisfy customers [17], and they usually set return policies by product category, rather than an individual product. It is commonly observed that return policies are announced ahead of the price. Thus, the return policy is usually a relatively long-term decision and is decided in advanced. Therefore, we assume that the manufacturer decides the return policy for direct channels and the retailer decides that for the indirect channel simultaneously in the first stage.

We consider that customers are heterogeneous in product valuation. In other words, consumers value the product that is from either channel at $v$, which is a uniform distribution between zero and one. This is similar to McWilliams [11]. If a consumer is satisfied with the product purchased, he/she will value the product at $v$ and keep it. Otherwise, the consumer retains the product valued at zero if the channel does not offer a return policy ($k_j = N$), or he/she returns it to the retailer (or manufacturer) and receives the full refund $p_j^K$ if the channel offers an MBG ($k_j = G$). Furthermore, we denote $\theta_j$ as the average customer satisfaction rate when a customer purchases by channel $j$. It is observed that $\theta_j$ relies on the offered service of channel $j$. According to the discussion above, the customer can more easily determine the product fit based on trial in the retail channel (i.e., channel $r$). However, in the direct channel, the information about products is captured only by others' reviews, except the official introduction. In practice, the online return rate is much higher than the off-line return rate, which means that the customer's satisfaction in channel $m$ is lower than in channel $r$. Therefore, it is reasonable for us to assume that $0 < \theta_m < \theta_r < 1$. Figure 1 shows the dual-channel structure, and the dotted lines are return channels, which exist only when the channel $j$ provides an MBG.

We assume that if the direct channel provides a return service, the returned products of the channel shall be handled by the manufacturer. Similarly, if a return service is provided through the indirect channel, the returned products are handled by the retailer. Let $c, s_j, T_j$ be the unit cost of producing products, the unit salvage of the returned product, and the unit cost of handling the returned product. Since the salvage and processing cost of the same product are almost the same, we make $s_r = s_m = s$, $T_r = T_m = T$. $t_j$ is the average of the unit cost for customers to return the product through channel $j$, meaning the transportation cost to return the product to a retail store or the mailing cost to return products purchased online. In practice, it is hard to distinguish the greater cost for customers that returned a product in the two channels. For the ease of exposition, we assume $t_r = t_m = t$ in the basic model. In line with Bintong Chen et al. [30], we define $E = s - t - T$ as the net salvage value of a returned product obtained by channel $j$. It is clearly found that $E$ reflects the channel's overall ability at

handling the returned product. Furthermore, to prevent the manufacturer and retailer from profiting by the salvage value of returned products, we assume that $s < c$.

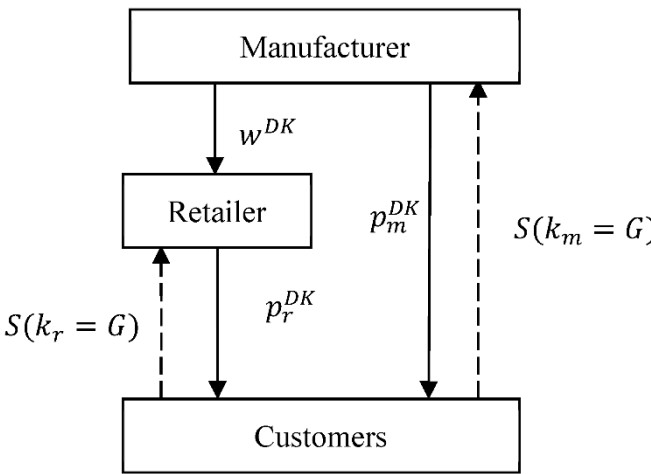

**Figure 1.** Dual supply chain model with return.

## 4. Analysis

We consider the four different return strategies, i.e., $K = k_m k_r = \{NN, NG, GN, GG\}$, and Figure 2 illustrates the sequence of events. First, the manufacturer decides the return policy (MBG or no MBG) for two channels simultaneously. Second, the manufacturer distributes products to the indirect channel at wholesale price $w^K$ and sells it through the online channel at price $p_m^K$. The retailer then sells the product through the indirect channel at retail price $p_r^K$. Third, customers make the channel choice by considering satisfaction rates, price, and the return policy. Therefore, we obtain the results as summarized in Proposition 1 (detailed derivations are shown in the proof in Appendix A). The superscript "∗" represents equilibrium.

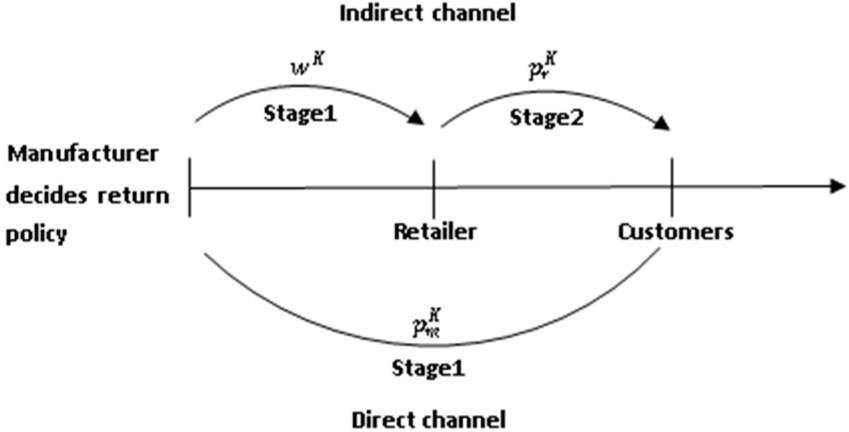

**Figure 2.** Sequence of events.

### 4.1. Third-Stage Game: Customer's Choice

The customer's choice of the channel depends on the satisfaction rate, return policy, retail price, and the cost of returning a product. Therefore, if a customer who values the product at $v$ considers purchasing the product through channel $j$, its utility is:

$$
U_j^K = \begin{cases} \theta_j v - p_j^K & k_j = N \\ \theta_j(v - p_j^K) - (1 - \theta_j)t & k_j = G \end{cases} \tag{1}
$$

Clearly, the customers will buy the product through channel $m$ if $U_m^K > U_r^K$ and $U_m^K > 0$, or they will choose the channel $r$ if $U_r^K > U_m^K$ and $U_r^K > 0$. Clearly, if $v \geq v_m^K$, the customers buy products through channel $m$. Meanwhile, they will change their choice to channel $r$ when $v \geq v_{mr}^K$, where $v_{mr}^K$ and $v_m^K$ can be derived from $U_r^K = U_m^K$ and $U_m^K = 0$ respectively. Then, the total customer demands for channels $m$ and $r$ are $D_m^K = v_{mr}^K - v_m^K$ and $D_r^K = 1 - v_{mr}^K$.

### 4.2. Second-Stage Decision: The Price Choice of the Retailer and Manufacturer

For given $K = k_m k_r = \{NN, NG, GN, GG\}$, the profit of the retailer is:

$$\Pi_r^K = \begin{cases} (p_r^{k_m N} - w^{k_m N}) D_r^{k_m N} & k_r = N \\ [\theta_r (p_r^{k_m G} - w^{k_m G}) + (1 - \theta_r)(s - T - w^{k_m G})] D_r^{k_m G} & k_r = G \end{cases} \quad (2)$$

The profit of the manufacturer is:

$$\Pi_m^K = \begin{cases} (w^{Nk_r} - c) D_r^{Nk_r} + (p_m^{Nk_r} - c) D_m^{Nk_r} & k_m = N \\ (w^{Gk_r} - c) D_r^{Gk_r} + [\theta_m (p_m^{Gk_r} - c) + (1 - \theta_m)(s - c - T)] D_m^{Gk_r} & k_m = G \end{cases} \quad (3)$$

To facilitate the analysis of the impact of return decisions on pricing, demand, and profit for manufacturers and retailers, we define $\Delta_j^{k_j}$, which is only affected by the return policy, to reflect the channel $j's$ efficiency at selling the product and handling the returned product.

$$\Delta_j^{k_j} = \begin{cases} \theta_j - c & k_j = N \\ \theta_j - c + E(1 - \theta_j) & k_j = G \end{cases} \quad (4)$$

**Proposition 1.** *When $\theta_m / (2\theta_r - \theta_m) < \rho^{DK} < 1$, for $K = k_m k_r = \{NN, GN, NG, GG\}$, there exists a unique optimal equilibrium of prices for two channels (summarized in Table 1).*

**Table 1.** Equilibrium and profits of the dual-channel (channel $r$ and $m$).

| Return Strategies | NN | GN | NG | GG |
|---|---|---|---|---|
| $w^*$ | $c + \frac{\Delta_r^N}{2}$ | $c + \frac{\Delta_r^N}{2}$ | $c + \frac{\Delta_r^G}{2}$ | $c + \frac{\Delta_r^G}{2}$ |
| $p_m^*$ | $c + \frac{\Delta_m^N}{2}$ | $t + 1 - \frac{\Delta_m^G + 2t}{2\theta_m}$ | $c + \frac{\Delta_m^N}{2}$ | $t + 1 - \frac{\Delta_m^G + 2t}{2\theta_m}$ |
| $p_r^*$ | $c + \frac{3\Delta_r^N - \Delta_m^N}{4}$ | $c + \frac{3\Delta_r^N - \Delta_m^G}{4}$ | $t + 1 - \frac{\Delta_r^G + \Delta_m^N + 4t}{4\theta_r}$ | $t + 1 - \frac{\Delta_r^G + \Delta_m^G + 4t}{4\theta_r}$ |
| $\Pi_r^*$ | $\frac{(\Delta^N - \Delta_m^N)^2}{16(\theta_r - \theta_m)}$ | $\frac{(\Delta^N - \Delta_m^G)^2}{16(\theta_r - \theta_m)}$ | $\frac{(\Delta_r^G - \Delta_m^N)^2}{16(\theta_r - \theta_m)}$ | $\frac{(\Delta_r^G - \Delta_m^G)^2}{16(\theta_r - \theta_m)}$ |
| $\Pi_m^*$ | $\frac{(\Delta_r^N - \Delta_m^N)^2}{8(\theta_r - \theta_m)} + \frac{(\Delta_m^N)^2}{4\theta_m}$ | $\frac{(\Delta_r^N - \Delta_m^G)^2}{8(\theta_r - \theta_m)} + \frac{(\Delta_m^N)^2}{4\theta_m}$ | $\frac{(\Delta_r^G - \Delta_m^N)^2}{8(\theta_r - \theta_m)} + \frac{(\Delta_m^N)^2}{4\theta_m}$ | $\frac{(\Delta_r^G - \Delta_m^G)^2}{8(\theta_r - \theta_m)} + \frac{(\Delta_m^N)^2}{4\theta_m}$ |

$\rho^K = \Delta_m^{km} / \Delta_r^{kr}$ represents the system efficiency of selling the product in channel $m$ in relation to indirect channel $j$. If $v_{mr}^K < v_m^K$, or equivalently $\rho^K < \theta_m / (2\theta_r - \theta_m)$, there is no demand in the direct channel. Similarly, if $v_{mr}^K > 1$ or $\rho^K > 1$, no demand will exist in the indirect channel. Only if $\theta_m / (2\theta_r - \theta_m) < \rho^K < 1$ can the two channels operate normally.

### 4.3. First-Stage Decision: Return Policy Decision for Each Channel

In this section, we analyze the return strategy for the situation in which both channels have sales, i.e., $\theta_m / (2\theta_r - \theta_m) < \rho^K < 1$. We will research how the manufacturer and retailer determine the customer return policy for each channel.

**Theorem 1.** *When $\theta_m / (2\theta_r - \theta_m) < \rho^K < 1$, for any return policy offered for the indirect channel $k_r = \{N, G\}$, $\Pi_m^{Gkr} > \Pi_m^{Nkr}$ if $E > 0$, while $\Pi_m^{Gkr} < \Pi_m^{Nkr}$ if $E < 0$.*

Theorem 1 shows that when the net salvage value of the returned product is positive, the manufacturer is willing to offer an MBG return policy for the online channel regardless of whether the indirect channel provides a return policy. That is to say, if $E > 0$, for the manufacturer, the optimal decision is offering the MBG return policy. When the net residual value is less than zero, the returned product does not bring positive benefits to the manufacturer. Therefore, they will not provide this service at this time. For the manufacturer, the net salvage value of returned products includes not only the salvage of returned products that rely on the nature of the product, but the cost to process the return and the cost to the customer to return the product to the manufacturer. In addition, since the model assumes a two-channel system, there is inevitably competition between the two channels. Although the two channels compete for market share, when the manufacturer determines its return policy, it should be based on whether the channel can save returned products. Thus, for manufacturers, the benefits of providing a return policy can mitigate the harm caused by competition between channels.

**Theorem 2.** *When* $\theta_m / (2\theta_r - \theta_m) < \rho^K < 1$ *, for the retailer,*

(a)  *if* $E > 0$, $\Pi_r^{NG} > \Pi_r^{NN} > \Pi_r^{GG} > \Pi_r^{GN}$ *and*
(b)  *if* $E < 0$, $\Pi_r^{NG} < \Pi_r^{NN} < \Pi_r^{GG} < \Pi_r^{GN}$.

The preference of the retailer for the four strategies behaves differently from the preference of the manufacturer, as shown in Theorem 2. $E > 0$ means that a return policy can bring benefits. Since the retailer only cares about the profit of indirect channels, when $E > 0$, retailers are more inclined to provide a return service. The result is quite clear because the retailer can improve customer utility, so increasing demand in the indirect channel through the MBG. However, the competition within the two channels is fierce, and the model assumes a manufacturer-led supply chain structure. As a follower, the retailer's interests will be harmed by the competition among channels when the direct channel provides a return policy to improve the demand of the channel. Competitive pressure from another channel outweighs the benefits of providing a return policy, resulting in retailers preferring the *NN* to *GG* policy. In addition, the retailer is more inclined to the NN policy than GG. Although the provision of the MBG return policy can increase channel demand, the demand growth of indirect channels is not as good as that of direct channels. That is to say, some customers of indirect channels are taken away after the two channels provide the return policy.

*4.4. Impact of the Return Strategy in the Dual-Channel Supply Chain*

In the basic case, supposed that $\theta_m / (2\theta_r - \theta_m) < \rho^K < 1$ holds and the retail price is set as given in Table 1, where the return strategies $K = k_m k_r = \{NN, GN, NG, GG\}$. The demand of each channel can be obtained. Comparing the corresponding prices and demand, the impacts on pricing and demand are summarized as follows.

**Proposition 2.** *When the manufacturer offers general products in the online channel, for any* $K = k_m k_r = \{NN, GN, NG, GG\}$ *and if* $\theta_m / (2\theta_r - \theta_m) < \rho^K < 1$, *then*

(a)  $p_m^{NG} = p_m^{NN}$, $p_m^{GG} = p_m^{GN}$ *and*
(b)  *If* $E > 0$, $p_r^{NN} > p_r^{GN}$, $p_r^{NG} > p_r^{GG}$. *If* $E < 0$, $p_r^{NN} < p_r^{GN}$, $p_r^{NG} < p_r^{GG}$.

Proposition 2 shows that, when the manufacturer offers an MBG in the direct channel, no matter whether another channel offers the MBG policy, the optimal price of the direct channel is fixed. In other words, the optimal prices for the direct channel are influenced by the return policy in this channel. This is because an MBG increases the customer's perceived utility of purchasing the product by channel $j$, which prompts the retailer to increase the sales price. However, the optimal prices for the indirect channel are varied with the return strategy of the direct channel. When E > 0, regardless of what kind of return policy is adopted by the indirect channel, for the retailer, the sales price will be higher when

the direct channels do not provide a return policy, even if the wholesale price is the same. The sales price decreases when the direct channel offers an MBG. This shows that retailers pay more attention to competition between channels. When direct channels provide the MBG service, increase customer value, and attract more customers, the retailer will take some steps to protect the interests of the indirect channel. Therefore, they increase the attractiveness of indirect channels to consumers by reducing sales prices, thereby reducing customer loss.

**Proposition 3.** *When the manufacturer offers general products in the online channel, for any* $K = k_m k_r = \{NN, GN, NG, GG\}$ *and* $\theta_m / (2\theta_r - \theta_m) < \rho^K < 1$,

(1) *If* E > 0, $q_m^{GN} > q_m^{GG} > q_m^{NN} > q_m^{NG}$ *and* $q_r^{NG} > q_r^{NN} > q_r^{GG} > q_r^{GN}$, *and*

(2) *If* E < 0, $q_m^{GN} < q_m^{GG} < q_m^{NN} < q_m^{NG}$ *and* $q_r^{NG} < q_r^{NN} < q_r^{GG} < q_r^{GN}$.

Proposition 3 shows that the return policies of the two channels affect each other's channel demand. Under the condition that the net salvage value of returned products is greater than zero, if the direct channel takes the MBG, the demand on this channel increases whether the indirect channel provides an MBG or not. However, the indirect channel providing the MBG will reduce the extent to which the demand of the indirect channel increases. In other words, if the manufacturer wants to increase the demand of the direct channel, it is better to implement the GN strategy. However, in indirect channels, the adoption of MBG does not necessarily increase channel demand. From Proposition 3, we can find that the demand of indirect channels depends more on the return strategy of direct channels. When direct channels do not provide an MBG, demand for indirect channels is always higher. If the MBG is provided through indirect channels at this time, the increase in demand can be further increased. Therefore, when manufacturers want to increase indirect channel requirements, it is best to choose the NG strategy. However, when the net salvage value of the returned product is negative, the order of demand is exactly the opposite.

In general, when the net salvage value of a returned product is positive, an MBG always increases the demand of the channel, and if the net value is negative, implementing an MBG is not a good option. Therefore, the manufacturer needs to consider not only the net value of returned products, but also the competitive impact of the return strategy of another channel when formulating the strategy. We set $\Pi^K$ as the total profit of the dual-channel supply chain under different return strategies.

**Proposition 4.** *When* $\theta_m / (2\theta_r - \theta_m) \leq \rho^K \leq 1$ *and* E > 0,

(1) *If* $\rho^{GG} + \rho^{NG} > 6\theta_m / (4\theta_r - \theta_m)$, $\Pi^{GG} > \Pi^{NG} > \Pi^{NN}$; *while* $\rho^{GG} + \rho^{NG} < 6\theta_m / (4\theta_r - \theta_m)$, $\Pi^{NG} > \Pi^{GG} > \Pi^{GN}$ *and*

(2) *If* $\rho^{GN} + \rho^{NN} > 6\theta_m / (4\theta_r - \theta_m)$, $\Pi^{GG} > \Pi^{GN} > \Pi^{NN}$; *while* $\rho^{GN} + \rho^{NN} < 6\theta_m / (4\theta_r - \theta_m)$, $\Pi^{GG} > \Pi^{GN} > \Pi^{NN}$.

From Proposition 4, we find that for entire supply chain, the optimal return strategy is either GG or NG. Under the conditions that E > 0, i.e., offer an MBG, will bring benefits to supply chain members, when $\rho^{DGG} + \rho^{DNG} > 6\theta_m / (4\theta_r - \theta_m)$, the best return policy choice is GG, and while $\rho^{DGG} + \rho^{DNG} < 6\theta_m / (4\theta_r - \theta_m)$, the strategy NG will obtain more profits. From the above, we know that the NG is the most profitable strategy for the retailer, while the GG for the manufacturer. As the sales capacity of online channels grows, GG can benefit the whole supply chain more than NG. At this time, the manufacturer will implement the GG return strategy and compensate the retailer for a certain amount of profits through constructs, so that the members of dual-channel supply chain will reach a Pareto optimality.

## 5. Implementation of Product Customization in the Online Channel

In this section, we assume that manufacturer can receive the information of customers' personal preference through their online channel. The manufacturer offers the right for customers to design the

appearance of the product. For example, the customer can upload their favorite pictures, which will be printed on a T-shirt, shoes, and other products, or choose their favorite styles and colors for products. For the ease of presentation in this paper, we call this sales channel the online customization channel ($j = o$).

The personalized products have some impact on our basic model. On the one hand, since customers take part in the production process and add personal preferences to the product, the satisfaction of customers who buy the product through this channel will be much higher than the indirect channel. We assume the average customer satisfaction rate when the product is purchased through channel $o$ is $\theta_o$ and $\theta_o > \theta_r$. On the other hand, the manufacturer will not offer the MBG return policy in the online customization channel. This is because each personalized product is specific and produced for the consumers who designed it. Therefore, in this situation, the return policy combinations of the whole supply chain are only two $OK = k_o k_r = \{ONN, ONG\}$. Furthermore, the online customization channel needs to collect customer's demand for production, which costs more than setting up an ordinary online channel. The extra construction cost is recorded as $Q$, and $\alpha$ is the average added cost of each customized product. The sequence of events occurs the same as in Section 4.

*5.1. Equilibrium Solution*

To compare the online customization channel, we assume that the manufacturer sells the ordinary products in the traditional sales channel (channel $r$) and sells the personalized products in channel $o$ at the same time. The sequence of events is the same as the basic model. If a customer who values the product at $v$ considers purchasing the product through channel $o$, its utility is $U_o^{OK} = \theta_o v - p_o^{OK}$. The utility of the customer buying the product through channel $r$ is:

$$U_r^{OK} = \begin{cases} \theta_r v - p_r^{ONN} & OK = ONN \\ \theta_r(v - p_r^{ONG}) - (1 - \theta_r)t & OK = ONG \end{cases}$$

Like the basic model, the customers will buy the product through channel $c$ if $U_o^{OK} > U_r^{OK}$ and $U_o^{OK} > 0$. Clearly, the total customer demands for channels $r$ and $c$ are easy to acquire. We use $\Delta_o = \theta_o - c - \alpha$ to simplify results, which reflects the online customization channel efficiency for selling products.

For given $OK = k_o k_r = \{ONN, ONG\}$, the profit of the retailer is:

$$\Pi_r^{OK} = \begin{cases} (p_r^{NN} - w^{NN})d_r^{NN} & k_r = N \\ [\theta_r(p_r^{NG} - w^{NG}) + (1 - \theta_r)(s - T - w^{NG})]d_r^{NG} & k_r = G \end{cases}$$

The profit of the manufacturer is:

$$\Pi_o^{OK} = \begin{cases} (w^{NN} - c)d_r^{NN} + (p_o^{NN} - c)d_o^{NN} - Q & k_r = N \\ (w^{NG} - c)d_r^{NG} + (p_o^{NG} - c)d_m^{NG} - Q & k_r = G \end{cases}$$

$\rho^{OK} = \Delta_r^{kr}/\Delta_o$ represents the system efficiency of selling the product in the indirect channel in relation to the customer-made channel. From Table 2 and the above analysis, it is obvious that the two channels can co-exist only if $\theta_r < \rho^{OK} < 1$. The demands and profits of the manufacturer and retailer are summarized in Table 2.

**Proposition 5.** *For given $OK = k_o k_r = \{NN, NG\}$ , there exists a unique optimal equilibrium of the prices for the two channels (summarized in Table 2).*

**Table 2.** Price equilibrium and expected profits when introducing the online customization in the dual-channel.

| K | NN | NG |
|---|---|---|
| $w^{O*}$ | $c + \frac{\Delta_r^N}{2}$ | $c + \frac{\Delta_r^G}{2}$ |
| $p_o^{O*}$ | $c + \alpha + \frac{\Delta_o}{2}$ | $c + \alpha + \frac{\Delta_o}{2}$ |
| $p_r^{O*}$ | $c + \frac{3\theta_o\Delta_r^N - \theta_r\Delta_o}{4\theta_o}$ | $t + 1 - \frac{\theta_o\Delta_r^G + \theta_r\Delta_o + 4t\theta_o}{4\theta_o\theta_r}$ |
| $\Pi_r^{O*}$ | $\frac{(\theta_o\Delta_r^N - \theta_r\Delta_o)^2}{16(\theta_o-\theta_r)\theta_o\theta_r}$ | $\frac{(\theta_o\Delta_r^G - \theta_r\Delta_o)^2}{16(\theta_o-\theta_r)\theta_o\theta_r}$ |
| $\Pi_m^{O*}$ | $\frac{(\theta_o\Delta_r^N)^2 - (\theta_r\Delta_o)^2}{8(\theta_o-\theta_r)\theta_o\theta_r} + \frac{\Delta_o(\Delta_o-\Delta_r^N)}{4(\theta_o-\theta_r)} - Q$ | $\frac{(\theta_o\Delta_r^G)^2 - (\theta_r\Delta_o)^2}{8(\theta_o-\theta_r)\theta_o\theta_r} + \frac{\Delta_o(\Delta_o-\Delta_r^G)}{4(\theta_o-\theta_r)} - Q$ |

*5.2. Channel Selection and the Impact of Personalized Customization*

In this section, we find the basic conditions for the manufacturer to determine whether to open the online customization channel. Notice that the conditions apply to cases with and without MBG in either channel.

**Proposition 6.** *For any $OK = k_ok_r = \{NN, NG\}$, $\rho^{OK} = \Delta_r^{kr}/\Delta_o$, if $\theta_r < \rho^{OK} < 1$, the manufacturer can sell through both the retail channel and the online customization channel.*

Proposition 6 gives the basic conditions by which the retailer should select an online customization channel according to the demand. It shows that the channel decision relies on the relative efficiency of selling products in either channel, i.e., $\rho^{OK} = \Delta_r^{kr}/\Delta_o$. If the efficiencies of selling the product in the two channels are relatively close ($\theta_r < \rho^{OK} < 1$), the manufacturer can open an online customization channel based on the traditional sales channel. Notice that the boundary value only depends on $\theta_r$, and the range of dual-channel establishment is narrower with the increase of $\theta_r$. The ratio $\rho^{OK}$ measures the relative selling efficiency. If the indirect channel offers an MBG, it becomes more efficient, as long as $E > 0$. Opening an online customization channel may not benefit the manufacturer in some cases, because the channel does not allow customers to return the product that does not fit.

We also study the equilibrium pricing strategy when manufacturers open online customization channels and analyze the optimal return strategy. Since the products produced by online customization channels have more customized costs, the sales price will definitely be higher than the sales price of ordinary online products, so we only discuss the changes in the retail price of indirect channels.

**Proposition 7.** *When $E > 0$, $\theta_r < \rho^{OK} < 1$ and $\theta_m/(2\theta_r - \theta_m) < \rho^K < 1$, we have: if $\frac{\Delta_m^{km}}{\Delta_o} < \frac{\theta_r}{\theta_o}$, $p_r^{kmN} > p_r^{ONN}$, $p_r^{kmG} > p_r^{ONG}$; while $\frac{\Delta_m^{km}}{\Delta_o} > \frac{\theta_r}{\theta_o}$, $p_r^{kmN} < p_r^{ONN}$, $p_r^{kmG} < p_r^{ONG}$.*

We can learn from Proposition 7 that when a manufacturer opens an online customization channel, the sales price of the channel increases due to the increase in cost. Meanwhile, the sales price of the indirect channel will also increase to some extent due to the influence of the online channel price. This shows that the establishment of online customization channels not only increases the direct channels' sales price, but also has an important impact on the overall price of the supply chain. However, when the selling efficiency of the online channels has no difference before and after offering the customization, the advantages of online customization in improving the price will not be reflected.

**Proposition 8.** *When $E > 0$ and $\theta_r < \rho^{OK} < 1$, we have: if $\left(\rho^{ONN} + \rho^{ONG}\right) > \frac{2\theta_r}{\theta_o}$, $\Pi_m^{ONG} > \Pi_m^{ONN}$, $\Pi_r^{ONG} > \Pi_r^{ONN}$; and if $\left(\rho^{ONN} + \rho^{ONG}\right) < \frac{2\theta_r}{\theta_o}$, $\Pi_m^{ONG} < \Pi_m^{ONN}$, $\Pi_r^{ONG} < \Pi_r^{ONN}$.*

Proposition 8 shows an interesting result. When manufacturers open online customization channels, the manufacturer and retailer have made a unified decision about the choice of indirect channel return strategies. According to Theorems 1 and 2, when the net salvage of the returned product is positive ($E > 0$), both the manufacturer and retailer are willing to provide a return policy for the channel to achieve a higher demand. However, the online customization channel does not provide an "inappropriate" return policy for consumers' personalized products. Therefore, in order to increase the overall demand, under certain conditions, members of the supply chain prefer to provide an MBG policy in indirect channels. It is worth noting that the main factor affecting the threshold conditions for providing a return policy is still consumer satisfaction. By analyzing the threshold, we can see that if the consumer satisfaction of indirect channels is closer to the satisfaction of online customization channels, then supply chain members will have greater willingness to implement an MBG for indirect channels.

In addition, we compared the profit of the manufacturer in two dual-channel models, the common dual-channel and the dual-channel that opens online customization, by numerical examples. We are particularly interested in how customer's satisfaction influences the channel selection strategies of the manufacturer. We set $\theta_o = 0.85$, $\theta_m = 0.65$, c = 0.3, $\alpha = 0.02$, Q = 0.03 and vary the parameters values in the following ranges: $\theta_r \in (0.65, 0.80)$. Since $E > 0$, *GG* is the best return decision for manufacturers under ordinary dual-channels. Therefore, we compare the profit changes of *GG*, *ONN*, and *OGG* in consumer satisfaction of indirect channels in Figure 2.

**Corollary 1.** *For manufacturer, $\theta_r^0$, $\theta_r^1$ exist by fixing the values of other parameters,*

(1)    *When $\theta_m < \theta_r < \theta_r^0$, it is beneficial for manufacturers to open online customization channels, regardless of the return policy adopted.*

(2)    *When $\theta_r^0 < \theta_r < \theta_r^1$, ONG is the optimal decision for the manufacturer.*

(3)    *When $\theta_r^1 < \theta_r < \theta_o$, maintaining an ordinary dual-channel structure and implementing MBG policies in both channels are the best decisions for manufacturers.*

Figure 3 illustrates the effects of varying $\theta_r$ on the profit of the manufacturer under different decisions. When the satisfaction of consumers shopping in the indirect channel tends to the satisfaction of ordinary online channels, i.e., the satisfaction of indirect channels is low, the improvement from opening the online customization is quite obvious. As $\theta_r$ increases, the advantages brought by online customization are significantly reduced. When $\theta_r$ exceeds $\theta_r^0$, it is necessary to provide a return service as only relying on the online customization cannot bring more profits. If $\theta_r$ exceeds $\theta_r^1$, since consumers are more satisfied with general products, manufacturers do not have to pay more to open online customization channels. Moreover, as the net salvage of returned products increases, $\theta_r = \theta_r^0$, $\theta_r = \theta_r^1$ move to left. This is because of the cost that the manufacturer paid to process returned products, so that the willingness to provide an MBG is increased.

Figure 3 demonstrates that opening online customization channels can increase customer satisfaction and channel demand and generate significant profit increases for manufacturers. However, when the satisfaction of indirect channels is close to the satisfaction of online customization channels, the advantages brought by the opening of the customization channel cannot make up for the cost.

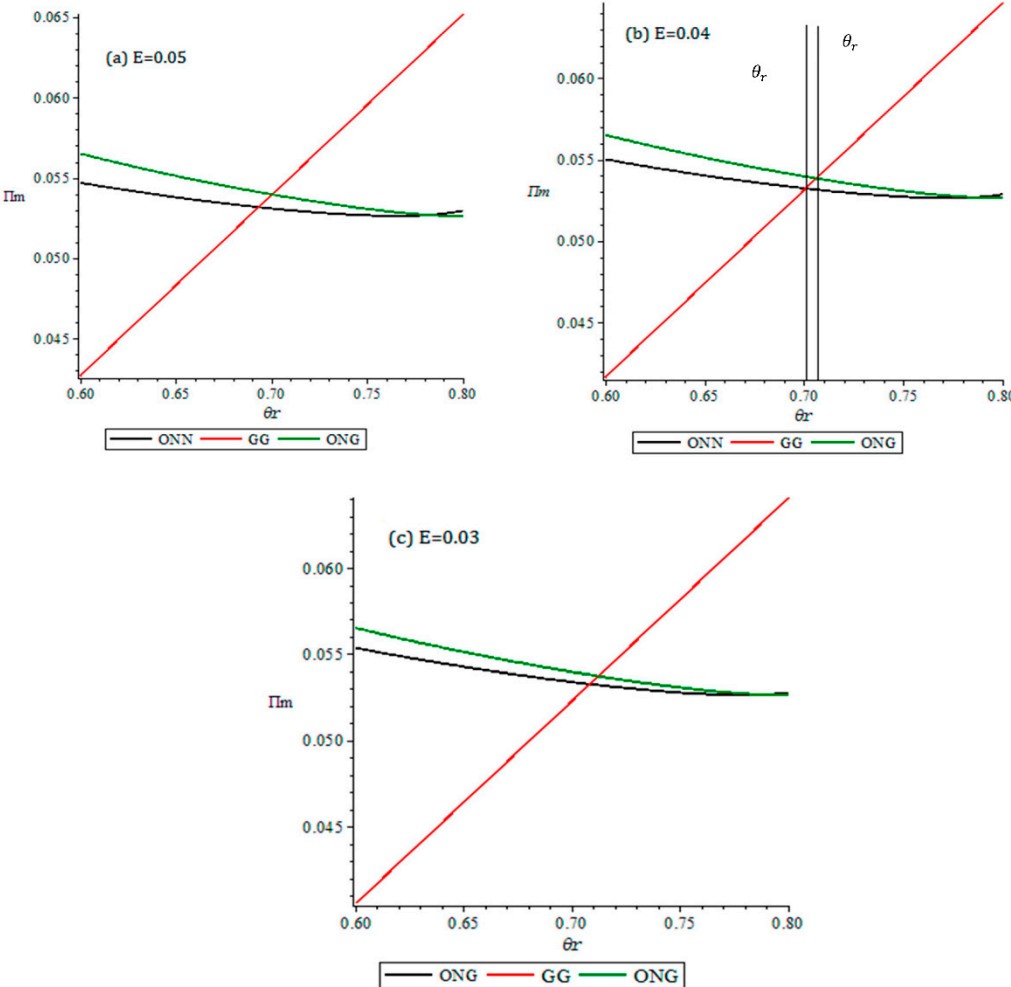

**Figure 3.** Effects of $\theta_r$ on the profits of the manufacturer under different *E*.

## 6. Conclusions

This study investigated the return strategies in a dual-channel supply chain composed of a manufacturer and a retailer and the conditions for the manufacturer to open online customization channels. In a single selling season, the manufacturer sells the product directly to the end customers and indirectly via a retailer. Specifically, two return policies were discussed: MBG and no MBG. Besides, from the perspective of manufacturers, we considered introducing product customization into online channels to increase customer satisfaction. Due to the uniqueness of customized products, online customization channels cannot provide the MBG policy. Therefore, we discussed the conditions under which manufacturers chose to open online customization channels.

We investigated how the manufacturer and retailer decide return policies. For the manufacturer, the decision on the return policy depended only on whether or not the channel was able to salvage the returned product, while the competition between channels would also have an impact. However, for a brick-and-mortar retailer, the opposite results occurred. For retailers, the impact of competition between channels is stronger, so that retailers are more willing to provide an MBG only in indirect channels. This has several implications. Besides, by comparing the total profits of supply chain channels, we found that there was a Pareto optimality between manufacturer and retailer, that is the two members could maximize the profits of the whole supply chain through agreements. We extended our discussion to the impacts of online customization on the manufacturer's channel selection and return strategy, when the manufacturer was capable of implementing customization in the online channel. We saw that the implementation of customization in the online channel may result in higher

sales prices in both channels. Meanwhile, when the manufacturer implemented customization in the online channel, the manufacturer and retailer unified the return policy for indirect channels. In addition, we compared the profit of the manufacturer before and after the online customization through numerical examples. We found that opening online customization channels did not always bring benefits to manufacturers. With the increase in consumer satisfaction in indirect channels, the implementation of online customization would hurt the manufacturer's profits.

We obtained some managerial implications from the study. First, the manufacturer should endeavor to increase the net salvage value of returned products through online channels by reducing the cost of processing customer returns or the cost of customer returns, so as to enhance the benefits of providing an MBG. Then, in channel selection, we believe that when a certain product is sensitive to customization, that is which consumers pay more attention to personalized characteristics, such as clothes, shoes, accessories, etc., the online personalized service can clearly bring benefits to manufacturers. For some products for which customers focus on function or do not need to reflect their personality, only providing a return service in the online channel is a better choice.

This work has a few limitations due to our model assumptions. First, this paper did not consider different consumer return costs in different channels. In practice, the cost of returning products through different channels is different, which will affect consumers' purchasing behavior, and thus affect channel demand. Second, this paper assumed that the manufacturer and retailer handle returned products in different channels. However, to save production costs, most of the returned products can be re-processed for second sales. In the future, we can consider the second sale of returned products. Finally, when considering opening online customization channels, we simply considered the basic characteristics of personalized products, especially in terms of cost. There are a number of customization channels that offer a no-reason return policy, and we can continue to research the return decision of customization channels in the future.

**Author Contributions:** Conceptualization, R.Z. and Z.H.; writing—original draft preparation, J.L.; writing—review and editing, R.Z. and B.L.; project administration, B.L.

**Funding:** National Science Foundation of China through Grant Number 71571117 and the Human and Social Science of Education Committee of China through Grant Number 18YJA630143.

**Conflicts of Interest:** The authors declare no conflict of interest.

**Appendix A**

**Proof of Proposition 1.** Since the analysis process for the dual-channel supply chain of different return policy combinations is similar, we take the NG case as an example to illustrate the solution process, with no MBG in the online channel and an MBG in the retail channel.

For K = NG, $U_m^{NG}(v) = \theta_m v - p_m^{NG}$ and $U_r^{DNG}(v) = \theta_r(v - p_r^{NG}) - (1 - \theta_m)t$. $v_m^{NG} = p_m^{NG}/\theta_m$, $v_{mr}^{NG} = \theta_r(p_r^{NG} - t) - (p_m^{NG} - t)/(\theta_r - \theta_m)$. Therefore, the demand of the two channels is $D_m^K = v_{mr}^K - v_m^K$ and $D_r^K = 1 - v_{mr}^K$, respectively.

For given $w^{NG}$, calculate the first and second derivatives of (2) on $p_r^{NG}$:

$\frac{\partial \prod_r^{NG}}{\partial p_r^{NG}} = \frac{\theta_r[-2\theta_r p_r^{NG} + p_m^{NG} + (s+t)(\theta_r - 1) + w^{NG} - \theta_m + \theta_r]}{\theta_r - \theta_m}$, $\frac{\partial^2 \prod_r^{NG}}{\partial p_r^{NG2}} = \frac{-2\theta_r^2}{\theta_r - \theta_m} < 0$.

A unique optimal $p_r^{NG}$ is:

$$p_r^{NG} = \frac{p_m^{NG} + (s+t)(\theta_r - 1) + w^{NG} - \theta_m + \theta_r}{2\theta_r}. \tag{A1}$$

Substituting (A1) into (3) and taking the first and second derivatives of (3) on $p_m^{NG}$ and $w^{NG}$:

$\frac{\partial \prod_m^{NG}}{\partial p_m^{NG}} = \frac{\theta_m(s-t)(1-\theta_r) + (2c+\theta_m)(\theta_m - \theta_r) - 2\theta_m(p_m^{NG} - w^{NG}) + 4\theta_r p_m^{NG}]}{2\theta_m(\theta_m - \theta_r)}$, $\frac{\partial^2 \prod_m^{NG}}{\partial p_m^{NG2}} = \frac{\theta_m - 2\theta_r}{\theta_m(\theta_r - \theta_m)} < 0$, and $\frac{\partial \prod_m^{NG}}{\partial w^{NG}} = \frac{(s-t)(1-\theta_r) + 2(p_m^{NG} - w^{NG}) - \theta_m + \theta_r}{2(\theta_r - \theta_m)}$, $\frac{\partial^2 \prod_m^{NG}}{\partial w^{NG2}} = \frac{-1}{\theta_r - \theta_m} < 0$.

Therefore, there exists a unique optimal pair ($p_m^{NG}$, $w^{NG}$):

$$p_m^{NG} = \frac{c + \theta_m}{2}, w^{NG} = \frac{c + \theta_r + (s - t)(1 - \theta_r)}{2}. \tag{A2}$$

Substituting (A2) into (A1), the indirect channel retail price is:

$$p_r^{NG} = \frac{3\theta_r + 2c - \theta_m - (1 - \theta_r)(s + 3t)}{4\theta_r}. \tag{A3}$$

The optimal prices for the other combinations are summarized in Table 1. □

**Proof of Theorem 1.** From Table 1, we compare the manufacturer's profits for the four return strategies. We have:

$$
\begin{aligned}
\Pi_m^{GG} - \Pi_m^{GN} &= \frac{E(1-\theta_r)[\Delta_r^G(1-\rho^{GG}) + \Delta_r^N(1-\rho^{GN})]}{8(\theta_r - \theta_m)}, \\
\Pi_m^{GG} - \Pi_m^{NG} &= \frac{E(1-\theta_m)\Delta_r^G[(2\theta_r - \theta_m)(\rho^{GG} + \rho^{NG}) - 2\theta_m]}{8\theta_m(\theta_r - \theta_m)}, \\
\Pi_m^{NG} - \Pi_m^{NN} &= \frac{E(1-\theta_r)[\Delta_r^G(1-\rho^{NG}) + \Delta_r^N(1-\rho^{NN})]}{8(\theta_r - \theta_m)}, \\
\Pi_m^{GN} - \Pi_m^{NN} &= \frac{E(1-\theta_m)\Delta_r^N[(2\theta_r - \theta_m)(\rho^{GN} + \rho^{NN}) - 2\theta_m]}{8\theta_m(\theta_r - \theta_m)}.
\end{aligned}
\tag{A4}
$$

If $E > 0$, we can derive the conclusion $\Pi_m^{GG} > \Pi_m^{GN} > \Pi_m^{NN}$ and $\Pi_m^{GG} > \Pi_m^{NG} > \Pi_m^{NN}$. If $E < 0$, $\Pi_m^{GG} < \Pi_m^{GN} < \Pi_m^{NN}$ and $\Pi_m^{GG} < \Pi_m^{NG} < \Pi_m^{NN}$. □

**Proof of Theorem 2.** From Table 1, we compare the retailer's profits for the four return strategies. We have:

$$
\begin{aligned}
\Pi_r^{NG} - \Pi_r^{NN} &= \frac{E(1-\theta_r)\Delta_m^N(\frac{1}{\rho^{NG}} + \frac{1}{\rho^{NN}} - 2)}{16(\theta_r - \theta_m)}, \\
\Pi_r^{NN} - \Pi_r^{GG} &= \frac{E(\theta_r - \theta_m)(2-E)}{16}, \\
\Pi_r^{GG} - \Pi_r^{GN} &= \frac{-E(1-\theta_m)\Delta_r^N(2 - \rho^{GN} - \rho^{NN})}{16(\theta_r - \theta_m)}.
\end{aligned}
\tag{A5}
$$

If $E > 0$, we can derive the conclusion $\Pi_r^{NG} > \Pi_r^{NN} > \Pi_r^{GG} > \Pi_r^{GN}$. If, $\Pi_r^{NG} < \Pi_r^{NN} < \Pi_r^{GG} < \Pi_r^{GN}$. □

**Proof of Proposition 2.** Comparing the prices of the dual-channel structure from Table 1, it is obvious that $p_m^{NG} = p_m^{NN}$, $p_m^{GG} = p_m^{GN}$.

$$p_r^{NN} - p_r^{GN} = \frac{E(1 - \theta_m)}{4}, p_r^{NG} - p_r^{GG} = \frac{E(1 - \theta_m)}{4\theta_r}. \tag{A6}$$

We have: when $E > 0$, $p_r^{NN} > p_r^{GN}$, $p_r^{NG} > p_r^{GG}$, while $E < 0$, $p_r^{NN} < p_r^{GN}$, $p_r^{NG} < p_r^{GG}$. □

**Proof of Proposition 3.** When $\theta_m/(2\theta_r - \theta_m) < \rho^K < 1$, we can obtain the demand of each channel, i.e., Table A1. □

**Table A1.** Demand of each channel.

| K | NN | GN | NG | GG |
|---|---|---|---|---|
| $q_m^K$ | $\frac{2\Delta_m^N - \lambda(\Delta_r^N + \Delta_m^N)}{4\lambda\theta_r(1-\lambda)}$ | $\frac{2\Delta_m^G - \lambda(\Delta_r^N + \Delta_m^G)}{4\lambda\theta_r(1-\lambda)}$ | $\frac{2\Delta_m^N - \lambda(\Delta_r^G + \Delta_m^N)}{4\lambda\theta_r(1-\lambda)}$ | $\frac{2\Delta_m^G - \lambda(\Delta_r^G + \Delta_m^G)}{4\lambda\theta_r(1-\lambda)}$ |
| $q_r^K$ | $\frac{\Delta_r^N - \Delta_m^N}{4(\theta_r - \theta_m)}$ | $\frac{\Delta_r^N - \Delta_m^G}{4(\theta_r - \theta_m)}$ | $\frac{\Delta_r^G - \Delta_m^N}{4(\theta_r - \theta_m)}$ | $\frac{\Delta_r^G - \Delta_m^G}{4(\theta_r - \theta_m)}$ |

From Table A1, we have:

$$q_m^{GN} - q_m^{GG} = \frac{\lambda(\Delta_r^G - \Delta_r^N)}{4\lambda\theta_r(1-\lambda)}, q_m^{GG} - q_m^{NN} = \frac{E(2-\theta_m)}{4\theta_m}, q_m^{NN} - q_m^{NG} = \frac{\lambda(\Delta_r^G - \Delta_r^N)}{4\lambda\theta_r(1-\lambda)},$$
$$q_r^{GN} - q_r^{GG} = \frac{\Delta_r^N - \Delta_r^G}{4(\theta_r - \theta_m)}, q_r^{GG} - q_r^{NN} = -\frac{E}{4}, q_r^{NN} - q_r^{NG} = \frac{\Delta_r^N - \Delta_r^G}{4(\theta_r - \theta_m)}. \tag{A7}$$

Therefore, when $E > 0$, we can derive the conclusion that $q_m^{GN} > q_m^{GG} > q_m^{NN} > q_m^{NG}$ and $q_r^{NG} > q_r^{NN} > q_r^{GG} > q_r^{GN}$. If $E < 0$, it turns out to be the opposite, i.e., $q_m^{GN} < q_m^{GG} < q_m^{NN} < q_m^{NG}$ and $q_r^{NG} < q_r^{NN} < q_r^{GG} < q_r^{GN}$.

**Proof of Proposition 4.** When $\theta_m / (2\theta_r - \theta_m) < \rho^K < 1$, we can obtain the total profits of the supply chain, i.e., Table A2. □

**Table A2.** Total profits of the whole supply chain.

| DK | NN | GN | NG | GG |
|---|---|---|---|---|
| $\Pi^{DK}$ | $\frac{3(\Delta_r^N - \Delta_m^N)^2}{16(\theta_r - \theta_m)} + \frac{(\Delta_m^N)^2}{4\theta_m}$ | $\frac{3(\Delta_r^N - \Delta_m^G)^2}{16(\theta_r - \theta_m)} + \frac{(\Delta_m^G)^2}{4\theta_m}$ | $\frac{3(\Delta_r^G - \Delta_m^N)^2}{16(\theta_r - \theta_m)} + \frac{(\Delta_m^N)^2}{4\theta_m}$ | $\frac{3(\Delta_r^G - \Delta_m^G)^2}{16(\theta_r - \theta_m)} + \frac{(\Delta_m^G)^2}{4\theta_m}$ |

From Table A2, we can get:

$$\Pi^{DGG} - \Pi^{DGN} = \frac{3E(1-\theta_r)[\Delta_r^G(1-\rho^{DGG}) + \Delta_r^N(1-\rho^{DGN})]}{16(\theta_r - \theta_m)},$$
$$\Pi^{DGN} - \Pi^{DNN} = \frac{E(1-\theta_m)\Delta_r^N[(4\theta_r - \theta_m)(\rho^{DGN} + \rho^{DNN}) - 6\theta_m]}{16\theta_m(\theta_r - \theta_m)},$$
$$\Pi^{DGG} - \Pi^{DNG} = \frac{E(1-\theta_m)\Delta_r^G[(4\theta_r - \theta_m)(\rho^{DGG} + \rho^{DNG}) - 6\theta_m]}{16\theta_m(\theta_r - \theta_m)}, \tag{A8}$$
$$\Pi^{DNG} - \Pi^{DNN} = \frac{3E(1-\theta_r)[\Delta_r^G(1-\rho^{DNG}) + \Delta_r^N(1-\rho^{DNN})]}{16(\theta_r - \theta_m)},$$
$$\Pi^{DGG} - \Pi^{DNN} = \frac{3(2-E)(\theta_r - \theta_m)}{16(\theta_r - \theta_m)} + \frac{E(1-\theta_m)(\Delta_m^G + \Delta_m^N)}{4\theta_m}.$$

Furthermore, we can have the following results.

When $E > 0$, $\Pi^{DGG} > \Pi^{DGN}$, $\Pi^{DNG} > \Pi^{DNN}$, $\Pi^{DGG} > \Pi^{DNN}$, $\rho^{DGG} + \rho^{DNG} > 6\theta_m / (4\theta_r - \theta_m)$, $\Pi^{DGG} > \Pi^{DNG}$, and $\rho^{DGN} + \rho^{DNN} > 6\theta_m / (4\theta_r - \theta_m)$, $\Pi^{DGN} > \Pi^{DNN}$.

**Proof of Proposition 5.** The proof is similar to the proof of Proposition 1. □

**Proof of Proposition 6.** If $v_{or}^{ONN} < v_r^{ONN}$ or equivalently $\rho^{OK} < \theta_r$, consumers will buy products through the online customization channel, which means there is no demand in the indirect channel. □

Similarly, if $v_{or}^{OK} > 1$, or $\rho^{OK} > 1$, no demand will exist in the online customization channel. Only when $\theta_r < \rho^{OK} < 1$ can the two channels operate normally.

**Proof of Proposition 7.** When $E > 0$, we have:

$$p_r^{NN} - p_r^{ONN} = \frac{\theta_r \Delta_o - \theta_o \Delta_m^N}{4\theta_o}, p_r^{GN} - p_r^{ONN} = \frac{\theta_r \Delta_o - \theta_o \Delta_m^G}{4\theta_o}. \tag{A9}$$

If $\frac{\Delta_m^{km}}{\Delta_o} < \frac{\theta_r}{\theta_o}$, $p_r^{kmN} > p_r^{ONN}$, while $\frac{\Delta_m^{km}}{\Delta_o} > \frac{\theta_r}{\theta_o}$, $p_r^{kmN} < p_r^{ONN}$.

$$p_r^{NG} - p_r^{ONG} = \frac{\theta_r \Delta_o - \theta_o \Delta_m^N}{4\theta_o}, p_r^{GG} - p_r^{ONN} = \frac{\theta_r \Delta_o - \theta_o \Delta_m^G}{4\theta_o}. \tag{A10}$$

If $\frac{\Delta_m^{km}}{\Delta_o} < \frac{\theta_r}{\theta_o}$, $p_r^{kmG} > p_r^{ONG}$, while $\frac{\Delta_m^{km}}{\Delta_o} > \frac{\theta_r}{\theta_o}$, $p_r^{kmG} < p_r^{ONG}$. □

**Proof of Proposition 8.**

$$\begin{aligned}
\Pi_m^{ONG} - \Pi_m^{ONN} &= \frac{E(1-\theta_r)(\theta_o\Delta_r^G + \theta_o\Delta_r^N - 2\theta_r\Delta_o)}{8\theta_r(\theta_o-\theta_r)}, \\
\Pi_r^{ONG} - \Pi_r^{ONN} &= \frac{E(1-\theta_r)(\theta_o\Delta_r^G + \theta_o\Delta_r^N - 2\theta_r\Delta_o)}{16\theta_r(\theta_o-\theta_r)}.
\end{aligned} \tag{A11}$$

When $E > 0$, we have: if $\left(\rho^{ONN} + \rho^{ONG}\right) > \frac{2\theta_r}{\theta_o}$, $\Pi_m^{ONG} > \Pi_m^{ONN}$, $\Pi_r^{ONG} > \Pi_r^{ONN}$; and if $\left(\rho^{ONN} + \rho^{ONG}\right) < \frac{2\theta_r}{\theta_o}$, $\Pi_m^{ONG} < \Pi_m^{ONN}$, $\Pi_r^{ONG} < \Pi_r^{ONN}$. □

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
