# Peer review of "Return Strategies and Online Product Customization in a Dual-Channel Supply Chain"

_sustainability, doi:10.3390/su11123482_

Round 1

Reviewer 1 Report

The paper gives a sound scientific analysis on the prospects of product personalisation in a dual supply channel. With some minor corrections, it can be an excellent research study.

Kindly try to shorten both the abstract and the conclusion. The results of the study given in the abstract should be explained in a precise and understandable way.

Kindly separate the implications of the study given in the conclusions under the heading managerial implications. It will also help in shortening the conclusions and would fetch more citations to the papers.

Kindly discuss how this topic is related to sustainable business in the introduction section. 

The authors could also mention a few sentences about the new strategy of personalised pricing.

Kindly correct the minor grammatical errors in the introduction and conclusion sections. 

Author Response

The paper gives a sound scientific analysis on the prospects of product personalisation in a dual supply channel. With some minor corrections, it can be an excellent research study.

1.      Kindly try to shorten both the abstract and the conclusion. The results of the study given in the abstract should be explained in a precise and understandable way.

Answer: We have modified the abstract and the conclusion as suggested, on pages 1 and 20-21 of the paper.

2.      Kindly separate the implications of the study given in the conclusions under the heading managerial implications. It will also help in shortening the conclusions and would fetch more citations to the papers.

Answer: We agree with the comments and improved this section on page 20 and 21 of the paper.

3.      Kindly discuss how this topic is related to sustainable business in the introduction section. 

Answer: The discussion is now defined on paragraph2, page 2 of the paper.

4.      The authors could also mention a few sentences about the new strategy of personalised pricing.

Answer: Personalized pricing is a powerful strategy for a retailer to provide products and services to meet the precise needs of a customer and to extract maximal surplus from that customer. Our research is focused on selling customized products rather than personalized pricing. Its pricing method is similar to that of ordinary products.

5.      Kindly correct the minor grammatical errors in the introduction and conclusion sections. 

Answer: We have carefully checked and corrected the paper.

Reviewer 2 Report

This work proposed the online product proposition channel in dual channel supply chain, and then examined the manufacturer’s channel selection and choice of returns policy in this case. The authors show that the online demand and profit of manufacturer will increase to a certain extent when open an online personalization channels. Still, compared to the case where both channels provide a Money-Back Guarantee (MBG), the implementation of online product personalization will damage manufacturer’s profits with the increase in consumer satisfaction in indirect channels. The manuscript is well crafted and authors did a great job in preparing the manuscript.  However, I have some comments which I would like to be addressed before the acceptance of the paper.

1.      Please describe the novelty of the present work in abstract.

2.      Why the optimal prices for indirect channel varied with the return strategy of the direct channel?

3.      What is the impact of MBG on direct and indirect channels Proposition 2?

4.      Please describe the relationship between Price equilibrium and expected profits of the dual-channel in Proposition 1?

5.      Please describe the relationship between sales price of the indirect channel and online channel price in Proposition 7?

6.      Please describe the managerial implications of your study?

Author Response

1.      Please describe the novelty of the present work in abstract.

Answer:First, we studied the return policy in dual-channel supply chain considering the competition of the two channels, and the demand of each channel relies on the satisfaction of customers. Second, considering the consumer satisfactions and the return strategy, the online customization channel is introduced into the manufacturer channel selection. That expanded the range of channel choices for manufacturers.

2.      Why the optimal prices for indirect channel varied with the return strategy of the direct channel?

Answer:In the dual-channel supply chain, there are competing relationships between the two channels. When the net salvage value E is positive, the direct channel makes itself more attractive by providing an MBG. Only reducing the sales price of the indirect channel can be recovered the lost customers under the condition that return policy unchanged. Therefore, the price of the indirect channel in the GN case will be lower than the price in the case of NN. Other price changes are similar to this.

3.      What is the impact of MBG on direct and indirect channels Proposition 2?

Answer:The sales price in the direct channel is related to whether the direct channel provides an MBG. However, for the sales price of the indirect channel, it is affected by the entire supply chain return strategy combination.

4.      Please describe the relationship between Price equilibrium and expected profits of the dual-channel in Proposition 1?

Answer:The expression in prosition1 is not accurate. It should be the optimal profit through the equilibrium price decision under different return policy combinations.

5.      Please describe the relationship between sales price of the indirect channel and online channel price in Proposition 7?

Answer:When online channels sell customized products, online channels cannot provide an MBG. For the price of indirect channels, when ratio of selling efficiency of the two type online channel is higher than the ratio of customer satisfaction of that two, the advantages of online customization in improving the sales price will be reflected.

6.      Please describe the managerial implications of your study?

Answer:We describe the managerial implications begins on line 12, page 21 of the paper.

Reviewer 3 Report

This paper considers return policy in a dual-channel supply chain. An upstream manufacturer has two channels to supply product: its own direct channel and an indirect channel (independent retailer). The direct channel offers lower quality service than the indirect channel. The quality levels are parameterized by \theta_m and \theta_r (the subscript represents manufacturer and (independent) retailer). The market structure is summarized in Figure 1 on page 8. Each channel determines its own return policy. If it allows unsatisfactory customers to return purchased goods, it needs to incur handing costs to salvage returned goods. The utility function of consumer with valuation v is summarized in equation (1) on page 9. Given the utility function, the model can be treated as a "vertical" differentiation model.

The timing of the page is as follows. First, the two channels simultaneously determine their own return policies (No guarantee and Guarantee). Thus, there are four subgames (NN, NG, GN, GG). Second, the manufacturer determines the retail price of its direct channel and the wholesale price for the independent retailer. Finally, observing the two prices, the independent retailer sets its price. The analytical results in the four subgames are summarized in Table 1. This is correct. Using the outcomes in the four subgames, this paper summarizes the equilibrium decisions of the two channels in Theorems 1 and 2. Each channel employs the return policy if and only if the net salvage value of a returned product (denoted by E) is positive.

This paper further investigates the properties of the outcome and extends the model (Section 5).

The outcomes are interesting. However, there are many typos, unsatisfactory statements, and expositional errors.

The first sentence in the first paragraph in Section 1 (page 2) is not acceptable in a competitive environment. Theoretically, vertical separation can be a credible commitment to aggressively set its amount of product (see, e.g., Saggi and Vettas (2002,Euro.Econ.Rev.). A proper sentence should be used.

The cited papers in Section 2 are not well crafted (but quite arbitrary "selected").

d_r and d_m in equations (2) and (3) would be D_r and D_m written in the end of Section 4.1.

The explanation in the paragraph right after equation 4 on page 10 should be written in the end of Section 4.1. This explanation is directly related to the condition that interior solutions exist. Also, in this paragraph, the second line would be "the direct channel m in relation to the indirect channel r". In addition, the superscript of \rho at the last equation would be K (not DK). This comment is applied to the first line in Proposition 1.

Around Proposition 4 on page 14, the superscripts have many typos. Please remove D. This kind of notational inconsistency exists in Proposition 5 and so on.

The above comments are not the full list of the expositional problem in this paper. Please carefully read and fix errors.  

Author Response

1.     The first sentence in the first paragraph in Section 1 (page 2) is not acceptable in a competitive environment. .......

Answer:The text has been changed as suggested, on page 2 of the paper.

2.      The cited papers in Section 2 are not well crafted (but quite arbitrary "selected").

Answer:We agree with the criticism. Improved text replacing this Section begins on page 5 of the paper.

3.      d_r and d_m in equations (2) and (3) would be D_r and D_m written in the end of Section 4.1.

Answer:The paper was corrected.

4.      The explanation in the paragraph right after equation 4 on page 10 should be written in the end of Section 4.1. .......

Answer:The suggested change was made, on pages 10 and 11 of the paper.

5.      Around Proposition 4 on page 14, the superscripts have many typos. Please remove D. This kind of notational inconsistency exists in Proposition 5 and so on.

Answer:The paper was corrected.

6.      The above comments are not the full list of the expositional problem in this paper. Please carefully read and fix errors. 

Answer:Done.

Round 2

Reviewer 2 Report

Authors have made all the required changes.